# Anxiety and Depression in Belgium during the First 15 Months of the COVID-19 Pandemic: A Longitudinal Study

**DOI:** 10.3390/bs12050141

**Published:** 2022-05-12

**Authors:** Helena Bruggeman, Pierre Smith, Finaba Berete, Stefaan Demarest, Lize Hermans, Elise Braekman, Rana Charafeddine, Sabine Drieskens, Karin De Ridder, Lydia Gisle

**Affiliations:** 1Sciensano, Epidemiology and Public Health, 1050 Brussels, Belgium; pierre.smith@sciensano.be (P.S.); finaba.berete@sciensano.be (F.B.); stefaan.demarest@sciensano.be (S.D.); lize.hermans@sciensano.be (L.H.); elise.braekman@sciensano.be (E.B.); rana.charafeddine@sciensano.be (R.C.); sabine.drieskens@sciensano.be (S.D.); karin.deridder@sciensano.be (K.D.R.); lydia.gisle@sciensano.be (L.G.); 2Institute of Health and Society, Université Catholique de Louvain, 1200 Brussels, Belgium

**Keywords:** COVID-19 pandemic, mental health, generalised anxiety disorder, depressive disorder, longitudinal study

## Abstract

The COVID-19 pandemic and policy measures enacted to contain the spread of the coronavirus have had nationwide psychological effects. This study aimed to assess the impact of the first 15 months of the COVID-19 pandemic on the level of anxiety (GAD-7 scale) and depression (PHQ-9 scale) of the Belgian adult population. A longitudinal study was conducted from April 2020 to June 2021, with 1838 respondents participating in 6 online surveys. Linear mixed models were used to model the associations between the predictor variables and the mental health outcomes. Results showed that the prevalence of symptoms of anxiety and depression was higher in times of stricter policy measures. Furthermore, after the initial stress from the outbreak, coping and adjustment were observed in participants, as symptoms of anxiety and depression decreased during times of lower policy restrictions to almost the same level as in pre-COVID times (2018). Though time trends were similar for all population subgroups, higher levels of both anxiety and depression were generally found among women, young people, people with poor social support, extraverts, people having pre-existing psychological problems, and people who were infected/exposed to the COVID-19 virus. Therefore, investment in mental health treatment programs and supports, especially for those risk groups, is crucial.

## 1. Introduction

Following the COVID-19 pandemic, substantial restrictions were introduced by governments, such as limiting the time allowed outside homes, initiating telework, closing childcare and school facilities, and turning to online education [1]. The measures with the most drastic social and economic impact—including closures of schools, bars, restaurants, and nightclubs, change in work situations, and restrictions on the number of people allowed to meet—were maintained over long periods of time despite the variability in how strict they were implemented [2].

Although effective in containing the spread of the virus [3], these added measures to the critical health situation can have unintended consequences for the mental health of the population. Besides the vast changes entailed in the social and personal life of people [4], other factors may trigger anxiety or depression, such as the unknown parameters and novelty of COVID-19, daily morbidity and mortality statistics in the news, concerns about the future [5], severe economic sanctions imposed on a country, and doubts about the adequacy of measures and the provision of health and medical services to control the disease.

Several studies assessed the psychological impact of the pandemic on the general population and highlighted a decrease in well-being, with higher scores on depression, anxiety, and stress, compared with baseline measures [6,7,8]. Moreover, the COVID-19 outbreak and subsequent policy measures may not affect all sociodemographic groups in the same way. Previous research suggests that young people might be disproportionally affected during the pandemic [9,10], for several reasons, such as decreased peer contact, decreased social activities, closure of schools, universities and support services, and increased pressure on families [11,12]. Apart from young people, other sociodemographic groups seem to be more vulnerable to mental health problems during the COVID-19 pandemic, such as people with lower socioeconomic status, women, single parents, and people with low social support [9,13,14,15,16]. Furthermore, having pre-existing mental health problems is associated with higher levels of psychological distress during this crisis [17]. Another risk factor related to the COVID-19 pandemic is having an infected relative [9,13,18].

Finally, some personality traits can also influence mental health in times of crisis. For example, changes in social life due to the COVID-19 restrictions can impact extravert and introvert people differently. In particular, introverts and extraverts exhibit fundamentally different approaches to social life [19,20], suggesting that the effect of social distancing might vary depending on individuals’ extraversion levels. It is possible that introverts are less adversely affected by physical distancing policies. As introverts typically have fewer social interactions than extraverts [21,22,23], physical distancing requirements might produce relatively small shifts in their behaviour, leaving their feelings of social connection unchanged. However, more research is needed to assess the association between individuals’ psychological traits and their mental health during the COVID-19 pandemic.

The main aim of this paper is to contribute to a better understanding of the short- and long-term effects of the COVID-19 pandemic and restriction policies on the mental health of the Belgian population during the COVID-19 pandemic. Understanding the impact of the COVID-19 pandemic is important, as other similar crises might occur in the future.

The following hypotheses are proposed:

(1)During the first 15 months of the pandemic in Belgium, times of strict lockdown measures are associated with a higher prevalence of symptoms of anxiety and depression, compared with times of relaxation of the measures;(2)The following risk factors are associated with anxiety and depression during the COVID-19 pandemic in Belgium: female gender, young age, single parent, extravert personality, exposure to COVID-19, lower level of education, and having a pre-existing mental health problem;(3)Time periods with stricter policy measures have greater impacts on mental distress among the high-risk groups mentioned above.

## 2. Materials and Methods

### 2.1. Setting

Belgium was hit severely by the COVID-19 pandemic. It was one of the first European countries to implement suppression and nationwide lockdown measures. The Belgian government took the first restriction measures on 13 March 2020 by announcing the closure of schools, bars, and restaurants. Five days later, a lockdown was declared; non-essential journeys and social gatherings were prohibited, and non-essential services and shops were closed. This period is designated as ‘severe restrictions’ in Table 1. By the end of May 2020, infection rates dropped, and the measures were gradually loosened in an exit strategy. Although COVID-19 still dominated social life, personal contact with other people was again possible during the summer of 2020, as was travelling in many European countries. The restrictions shifted from ‘moderate’ to ‘low’ (Table 1). However, starting from September 2020, the number of infections, hospital admissions, and deaths due to COVID-19 increased again, and by the beginning of November 2020, stronger restrictions in social and professional life were implemented. Christmas and New Year were celebrated under limiting but ‘moderate restrictions’. These restrictions continued until the end of April 2021, with fluctuating restrictions regarding non-essential activities, possibilities to travel, the opening of shops, amount of social contacts allowed, obligations of teleworking, etc. From the beginning of May 2021, there was again a period of relaxation of these measures, for example, with the reopening of schools, bars, and restaurants.

### 2.2. Dataset

A series of online COVID-19 health surveys, organised by Sciensano, the Belgian Institute of Public Health, was launched to evaluate the impact of the COVID-19 crisis on people’s daily life. This study used data from 6 different surveys (waves) organised in the first 15 months of the COVID-19 pandemic (from April 2020 to June 2021). Table 1 shows the exact dates of the data collection waves, together with the number of participants and the level of restrictions imposed by the government at each time point. Only Belgian residents (as well as foreigners living in Belgium) aged 18 years or older who gave their consent were allowed to take part in the survey [24]. All the surveys were developed in LimeSurvey version 3.

The link to the survey was announced on the website of Sciensano, as well as on the website of other organisations, press, and social media websites. In addition, starting from the second wave, all participants in the previous COVID-19 health surveys who had shown their willingness to participate in a subsequent survey (and had provided their e-mail address for this purpose) were invited by e-mail to take part in the following COVID-19 health survey.

This study focussed on respondents who participated in all six survey waves, providing full longitudinal data profiles (N = 1838).

To allow a comparison with the level of mental distress found in the Belgian population under normal conditions, the data of the Belgian Health Interview Survey of 2018 (*n*  =  7793) were used, hereafter named BHIS2018 [25,26]. This survey, which has been carried out every 4–5 years since 1997, assesses the health status of the Belgian population and various social and behavioural determinants.

As the composition of the two samples (Appendix A) were biased, the prevalence rates of symptoms of anxiety and depression were corrected based on post-stratification weights, and the models were fit conditioning on age, gender, level of education, and employment status (see Section 2.3.3. Statistical Analysis).

### 2.3. Measures

#### 2.3.1. Mental Distress

In this study, two dimensions of mental health that could be the most impacted by the COVID-19 crisis were selected: general anxiety and depressive disorder. These outcome variables were measured in all 6 waves and in the BHIS2018 survey, which enabled the comparison with pre-COVID times.

For generalised anxiety disorders, the total sum score of the 7-item Generalised Anxiety Disorder questionnaire (GAD-7) [27] was used (see Appendix B). One can answer these questions on a scale from 0 = ‘not at all’ to 3 = ‘almost every day’, with a reference period of the past 2 weeks. The total score between 0 and 21 is dichotomised at the cutoff value of 10+ for case definition. The specificity, sensitivity, and internal consistency of this scale are greater than 0.8 [27].

For depressive disorders (including major depressive disorder and other forms of depression), the sum score (range 0 to 27) based on the Patient Health Questionnaire, PHQ-9 [28] was used (see Appendix B). The questionnaire scores the nine DSM-IV criteria for depression on a 4-point scale from 0 (not present) to 3 (present almost every day). In order to create a binary variable, people were indicated with a depressive disorder when they have (1) a score > 1 (present more than half the time) on at least five items, where the first two items (‘little interest or pleasure in activities’ and ‘feeling dejected, depressed or despondent’) should be scored, and where a score > 0 is already sufficient on the item ‘feeling down, depressed or despondent’, or (2) when people’s score on the two first items (score >0) in combination with a score > 2 on 2–4 other items, where a score >0 is already sufficient for the item ‘thoughts that you would be better off if you were no longer alive’.

This scale also uses the reference period of the last 2 weeks and has reliable specificity, sensitivity, and internal consistency [28].

#### 2.3.2. Predictor Variables

Based on the literature on mental health during the COVID-19 pandemic [8,29] and the hypothesis developed in the introduction section, the following predictor variables were included in the analysis.

Demographic characteristics included age, gender, region of residence, employment status, educational level, and living situation (household composition).

Pre-existing mental health problems were measured by a proxy in the first survey wave: whether people reported having had an appointment with a psychologist, psychotherapist, or psychiatrist in the 4 weeks preceding the pandemic.

To know if someone was infected/exposed to COVID-19, respondents were asked at wave 5 if they had a current or past COVID-19 infection. Moreover, in wave 2, respondents were asked if they had a household member, family member, friend, or colleague who has a current or past COVID-19 infection.

Social support was measured in all 6 waves using the 3-item Oslo Social Support Scale (OSSS-3) [30]. A total score of 3–8 reflects ‘poor support’, a score between 9 and 11 indicates ‘moderate support’, and a score between 12 and 14 stands for ‘strong support’.

The personality trait extraversion was measured by the 6-item extraversion subscale of the Big Five Inventory-2 Short Form (BFI-2-S) [31]. This Short Form was used instead of the full BFI-2 scale in order to limit the non-response rate [32]. Participants rated their agreement with statements such as ‘I am someone who is outgoing, social’ and ‘I am someone who is shy, introverted’ (scored from 1—strongly disagree to 5—strongly agree). A mean score smaller than 2 is an indication of ‘introversion’, a score between 2 and 3 indicates a ‘middle’ score, and a score higher than 3 indicates ‘extraversion’. The BFI-2-S provides approximately 10% less reliability and validity than the full BFI-2 domain scales. However, an adequate statistical power could be remained in this study because of the large sample of participants [33].

#### 2.3.3. Statistical Analysis

For the descriptive analysis of the prevalence of anxiety and depression, the dichotomised variables for anxiety and depression were used. Due to a difference in our study composition and the composition of the general population (18 years or older) in terms of gender, age group and educational level, the unweighted probability of the outcome variables (anxiety and depression) would be biased. To compensate for this, we presented the weighted dichotomised outcome variables, based on post-stratification weights [24] by taking into account gender, age group, province, and educational level.

Continuous outcome variables (total score on the GAD-7 and PHQ-9 scales) were used to model the association between the predictor variables and mental health outcomes because of a reduction in statistical power to detect the relationship between mental health outcomes and predictor variables [34], using a dichotomised outcome variable. Random effects were used to allow different starting points, implying a specific model for each individual. The linear mixed model can be found in Appendix C.

#### 2.3.4. Model Building

In order to answer the research hypotheses posed in this study, we started with a null or empty model that reports whether there is a significant variance between individuals regarding anxiety and depression.

The second step of the modelling process consisted of including the second-level time variable, together with a random intercept, depending on its significance. The third model included the time variable (with the third time period as a reference period because this represents a time period with low restrictions) and six first-level predictor variables—namely, gender, age categories, region of residence, employment status, educational level, and household composition. In the fourth model, we added the other variables of interest: level of social support, personality trait extraversion, appointment with a psychologist/psychiatrist, and exposure to COVID-19. As the last step, the interaction effects between time and all other predictor variables were included one by one in the model. Only the models with a significant interaction effect were reported.

Regarding the fit of the models, O’Connell and McCoach [35] stated that differences of 2–5 between the model fit statistics (AIC or BIC) provide positive evidence for favouring the more complex model, differences of 6–10 provide strong evidence, and differences above 10 provide very strong evidence for favouring the more complex model.

A missing values analysis with Little’s test [36] showed that data were missing completely at random (Little’s MCAR test, Chi-Square (401) = 404.792; *p* = 0.44). To ensure comparability of the models in terms of completeness, missing variables were imputed using the fully conditional specification method. The proportion of missing values ranged from 0.5% to 6.5% (Appendix A). For the binary and ordinary variables with missing values, a logistic regression method was used to impute missing values. For the continuous variables with missing values, the Mean Matching method was used [37]. We created 5 imputed datasets. This number was enough to achieve a very good efficiency (the relative efficiency was close to 1.0 for all effects).

## 3. Results

### 3.1. Descriptive Statistics

Table 2 shows the demographic sample characteristics and descriptive data of all predictor variables. A total of 1838 respondents were included in the sample, aged 20–88, with a mean age of 53.5 years (SD = 13.5), during the first wave (T1). Furthermore, the majority of the participants were female (63.2%), lived as a couple without children (40.7%), were in a paid job (64.2%), and had a higher degree of education (74.0%). Some of these variables have changed over time within individuals, resulting in a slight variation in figures over time, as shown in Table 2. Furthermore, 74.2% had been infected/exposed to the COVID-19 virus (4.0% reported infection with the COVID-19 virus at wave 6 and 72.9% reported having an infected household member, family member, friend, or colleague at wave 2). In terms of the personality trait extraversion, 12.9% scored as introverts, 70.4% in the middle, and 16.7% as extraverts. Lastly, 10.2% of the respondents had a planned consultation with a psychologist/psychotherapist or psychiatrist the last 4 weeks before the start of the pandemic.

Frequency distributions of both anxiety and depression during the COVID-19 crisis are presented in Figure 1. During the pandemic, the weighted prevalence of anxiety and depression were higher in all waves, compared with the anxiety and depression rates during pre-COVID-19 times (HIS2018) (11.2% and 9.4%). Furthermore, during times of stricter restrictions (T1, 4, and 5), the rates of anxiety and depression were the highest (Figure 1).

### 3.2. Predictors of Mental Distress

#### 3.2.1. Anxiety

Results from linear mixed models showed the total variability in the continuous anxiety score (GAD-7 scale). In each model (Table 3), the variability was decomposed into two parts: the variability between the outcomes within every respondent (level 1 error variance) and the variability between respondents (level 2 error variance).

First, the null model showed that the average anxiety score equalled 4.16 (SD = 1.00, *p <* 0.001). The findings from the null model (model 1), with random intercept only, indicated that there was significant variation between individuals regarding the anxiety score. The null model indicated that 65.6% (14.7/(14.7 + 7.7)) of the total unexplained variance in the outcome variable anxiety (intraclass correlation (ICC)) was attributed to differences between individuals, which provides evidence for using multilevel regression models. As seen, this unexplained variance decreased as the models became more complex (60.5% in model 5).

As the models in this study became more complex, the AIC and BIC values (model fit statistics) decreased between the null model and the full models (>10, see Section 2. Materials and Methods), thus indicating very strong evidence for a better model fit throughout the progression of models.

The final model (model 5) showed that there was a significant effect of time on the expected anxiety score. One can see a significant higher anxiety score (0.75, SD = 0.15, *p* < 0.001) during T1 (severe restrictions), compared with T3 (low restrictions). The anxiety score was also estimated to be significantly higher during T4 (0.35, SD = 0.15, *p* < 0.05) and T5 (0.33, SD = 0.15, *p* < 0.05) (both moderate restrictions), compared with T3. The score on anxiety decreased in the last time period of June 2021 (low restrictions), compared with T3 (−0.23, SD = 0.15), although this difference was not statistically significant (*p* > 0.05).

Model 5 showed that the following groups were significantly more likely to have higher levels of anxiety: Women (ref. = men, 0.69, SD = 0.21, *p* < 0.001), young people (18–29 years old) (ref. = 65+, 1.32, SD = 0.39, *p* < 0.01), people aged 30–49 year (ref. = 65+, 1.08, SD = 0.26, *p* < 0.001), and people with a lower degree of education (ref. = higher degree, 0.44, SD = 0.17, *p* < 0.01). Furthermore, unemployed people (ref. = paid job, 0.70, SD = 0.32, *p* < 0.05) were more likely to have higher levels of anxiety. The opposite trend was shown for retired people (ref. = paid job, −0.68, SD = 0.20, *p* < 0.001).

Furthermore, the results of model 5 showed that the following groups were more likely to have higher levels of anxiety: people had previously planned an appointment with a psychologist/psychotherapist or psychiatrist (ref.= people who had not, 2.60, SD = 0.29, *p* < 0.001), extravert people (ref. = introvert people, 1.98, SD = 0.32, *p* < 0.001), people who were infected/exposed to the COVID-19 virus (ref.= no infection/exposure to COVID-19, 0.47, SD = 0.19, *p* < 0.01), and people who reported poor (ref. = strong social support, 1.32, SD = 0.12, *p* < 0.001) to moderate (ref. = strong social support, 0.44, SD = 0.09, *p* < 0.001) social support.

Besides these main effects, model 5 showed a significant interaction effect of time and gender. More specifically, the difference between T1 and T3 in anxiety was even larger for women (0.57, SD = 0.19, *p* < 0.01), compared with that for men. Other models that modelled the interaction effects of time and the other predictor variables (one by one) did not show significant interaction effects, and therefore, they were not listed in Table 3.

#### 3.2.2. Depression

Table 4 shows the linear mixed models for the continuous depression score (PHQ-9 scale). The first model (null model) showed that the average depression score is 4.18 (SD = 0.10, *p* < 0.001). Furthermore, the random intercept was significant (*p* < 0.001), meaning that there was variation between individuals’ average depression scores. The null model indicated an intraclass correlation (ICC) of 72% (17.34/(17.34 + 6.76)). This means that the difference between respondents’ depression scores explains 72% of the total variability, which can also be interpreted as the correlation between the scores of each individual. In these models, the unexplained variance also tended to decrease by adding more variables (to 66% in model 4).

The fit of the models became better (AIC and BIC values decreased) by adding more explanatory variables in the models (difference >10 from models 1 to 4). The model fit, however, did not improve after including the interaction terms between time and other variables. The interaction of model 5 with the variables is, thus, not shown in Table 4.

The final model (model 4) indicated a significant effect of time on the depression score. As for anxiety, we noted significant higher levels of depression at T1 (0.38, SD = 0.09, *p* < 0.001), T4 (0.62, SD = 0.09, *p* < 0.001) and T5 (0.68, SD = 0.09, *p* < 0.001), compared with T3.

Model 4 showed that the following groups were significantly more likely to have higher levels of depression: women (ref. = men, 0.49, SD = 0.18, *p* < 0.01), unemployed people (ref. = paid job, 0.72, SD = 0.31, *p* < 0.05), people on invalidity (ref. = paid job, 0.93, SD = 0.38, *p* < 0.001), young people (18–29 years old) (ref. = 65+, 1.61, SD = 0.38, *p* < 0.001), people aged 30–49 years (ref. = 65+, 1.11, SD = 0.26, *p* < *0*.001), and people aged 50–64 years (ref. = 65+, 0.51, SD = 0.21, *p* < 0.05). The opposite trend (more likely to have lower levels of depression) was shown for retired people (ref. = paid job, −0.64, SD = 0.19, *p* < 0.01).

Again, the same conclusions as for anxiety could be made—namely, that the following groups were more likely to have higher levels of depression: people who had previously planned an appointment with a psychologist/psychotherapist or psychiatrist (ref.= people who had not, 2.95, SD = 0.30, *p* < 0.001), extravert people and people who had a middle score on the extraversion scale (ref. = introvert people, 2.71, SD = 0.33, *p* < 0.001 and 0.75, SD = 0.27, *p <* 0.01, respectively), people who were infected/exposed to the COVID-19 virus (ref.= no contact with COVID-19, 0.43, SD = 0.21, *p* < 0.05), and people who reported poor (ref.= 1.49, SD = 0.11, *p* < 0.001) to moderate (ref. = strong social support, 0.50, SD = 0.09, *p* < 0.001) social support.

## 4. Discussion

In this longitudinal, population-based study, we tracked changes in levels of anxiety and depression during the first 15 months of the COVID-19 pandemic in Belgium and the associated factors.

This study highlighted that during the COVID-19 pandemic, the prevalence of symptoms of anxiety and depression evolved according to the degree of severity of the restriction policies (research hypothesis 1). The results of this study showed that, at the beginning of the crisis, a time when stay-at-home orders had been in place, the prevalence of symptoms of anxiety and depression was also significantly higher than the next time periods during the COVID-19 crisis (May 2020 and September 2020), characterised by less strict measures such as more possibilities to have social contacts, the reopening of schools and bars/restaurants, etc.

This provides some evidence of coping and adjustment after the initial stress triggered by the pandemic and subsequent strict lockdown measures. The evolution in mental health according to the evolution of the policy measures has also been observed in other Belgian studies [6,38,39]. The study of Beutels and Pepermans [38] also showed that mental health problems often occur before restrictions are announced, anticipating and probably as a result of changing perceived risks, and media coverage of impending changes. Furthermore, large and persistent psychosocial impacts of the COVID-19 among adults were found in a very diverse set of countries (in terms of epidemiological situations and sociocultural backgrounds) all over the world [40]. The COVID-19 pandemic has resulted in substantial global mental health challenges, such as increased levels of anxiety and depression symptoms [41,42]. A systematic review in 204 different countries [43] showed the increased prevalence of major depressive disorders and anxiety disorders.

Besides assessing the evolution of symptoms of anxiety and depression during the pandemic, we examined the differences in risk in population subgroups (research hypothesis 2). Although the prevalence of mental health problems was higher among all sociodemographic groups during times of stronger restrictions, our findings showed that women, young people, people with poor social support, extravert people, people being infected or exposed to the COVID-19 virus, and people with pre-existing mental health problems had higher levels of both anxiety and depression. In addition, having a lower educational level was associated with a higher risk of anxiety, while being unemployed was associated with a higher level of depression, compared with people with a paid job.

The finding that women were at greater risk of higher levels of anxiety and depression than men is consistent with previous studies [10,18,40,41]. Various hypotheses have been proposed to explain this increased vulnerability in women, including biological factors such as physiological reactivity and hormones. [44]. Other studies showed that gender is a better predictor than biological factors for explaining the difference in anxiety [45,46] and depression [47].

Regarding the third research question, we can state that, in line with overall trends, almost every subgroup experienced similar increases and decreases in anxiety and depression during the first 15 months of the COVID-19 pandemic. However, women were, in general, more at risk for higher levels of anxiety. This difference in risk was the largest at the beginning of the crisis (lockdown 1), which was not the case for the levels of depression, where the difference in gender remained constant over the different waves.

The longitudinal study of Daly et al. [1] had already suggested that women in the UK showed particularly pronounced declines in mental health status during the first lockdown in April 2020. One possible explanation can be that women may be experiencing a disproportional burden of the economic shock associated with the COVID-19 crisis during the first lockdown. For example, mothers in two-parent households have experienced greater increases in childcare responsibilities, interruptions to paid work, and job loss, compared with fathers in such households. [48] Another possible explanation can be that women work more often in the healthcare sector, which was highly confronted with the negative consequences of the COVID-19 virus and insecurities during the first lockdown. Studies of healthcare workers in China during the peak of the COVID-19 outbreak reported that those who worked in medical units and those who worked as front-liners had a high risk of exposure to COVID-19 patients and fear of being infected [49,50,51,52].

The downward trend of psychological distress according to age is a much-debated topic in the literature. A greater risk of psychological distress among young people was also found in an international study [40] that compared eight countries across four different continents, which was also confirmed in a Chinese study [53]. However, these findings are quite surprising given that older people are at greater mortality risk from COVID-19. One possible explanation can be that confinement measures have a particularly strong (social, occupational, and psychological) impact on younger people. As older people have, in general, a lower level of social, as well as professional activities, compared with younger people, the confinement measures might have less impacted their social lives [6,54]. While the use of digital technologies might mitigate some of the negative effects of social distancing, young people’s affinity with social media might also pose a threat to their well-being when they are confronted with information overload and ‘fake news’, which is especially detrimental during global crises [18,55].

Moreover, we expected that people with children [9,14] and people who are living alone [56] would be highly affected by the pandemic. However, both groups were not found as a significant predictor of higher levels of anxiety or depression in this study.

In contrast, the finding that a greater risk of higher levels of depression was associated with being unemployed (compared with people with a paid job) is consistent with the results of some other studies [57,58,59].

Besides financial insecurity, there are many other psychological costs of unemployment, including one’s potential loss of meaning in life, impairment of personal identity, and loss of the self-esteem that one typically draws from one’s job [60].

As expected, this study showed that people with poor and even moderate social support experience higher levels of anxiety and depression. Other research already showed that social support was negatively associated with mental health outcomes [61]. Social networks can give the opportunity to share feelings; thus, one can express fears and be better equipped at managing these feelings of fear [62]. Moreover, seeking social support is considered one of the coping strategies that may help overcome the fear related to the outbreak of an epidemic [63]. Furthermore, social support plays a crucial role in decreasing fear, as it enhances self-worth, social self-confidence, and the feeling that one can control the outcome of events in life [64].

Another risk factor associated with anxiety and depression that is uniquely linked to the COVID-19 situation is being infected by COVID-19 or having an infected relative. Consistent findings were found with other studies [9,13,18,65,66], according to which being infected/exposed to the COVID-19 virus was a significant predictor of mental distress.

In our multilevel mixed model, people with a pre-existing appointment with a psychologist or psychiatrist were at a significantly higher risk for both symptoms of anxiety and depression. However, in the literature, no evidence was found that there was a stronger increase in symptoms of anxiety and depression during the COVID-19 pandemic in those with a higher burden of mental disorders [67]. This study also provides no evidence of a higher increase in the level of anxiety and depression during times of COVID-19 since there was no significant interaction effect between time and having a pre-existing mental health problem.

Finally, the findings indicate that the levels of anxiety and depression are higher during times of COVID-19 among extravert people, compared with introvert people. This is in contrast with findings that, under normal conditions, extraversion is robustly associated with higher subjective well-being [68,69]. One possible explanation of these findings is that physical distancing requirements might produce relatively small shifts in introverts’ behaviour, leaving their feelings of social connection unchanged, because they typically have fewer social interactions than extraverts [21,22,23]. Other studies also showed that COVID-19 measures feel more unnatural for extravert people, leading to an increasing level of mental distress and the feeling of social disconnectedness, compared with introvert people [70,71]. However, no evidence was found of a different evolution of mental distress during the different periods of the COVID-19 pandemic between extravert or introvert people.

### Strengths and Limitations

This study contributes to existing research in important ways: first, this study sheds light on changes in the level of anxiety and depression, at the population level, associated with the pandemic and the accompanying prevention measures.

Due to performing a longitudinal study (with six measurement points), we can ensure that the declines in levels of anxiety and depression could not be attributed to differences in the sample across time points. Moreover, the large sample provided sufficient power to estimate the patterns of change in levels of anxiety and depression across population subgroups including those clinically vulnerable to COVID-19. This study contained a whole range of possible predictive factors, which contributes to an understanding of which specific groups of people are at higher risk to have higher levels of anxiety and depression.

Another strength of this research is that we used well-validated screening tools (GAD-7 and PHQ-9) to assess the incidence of anxiety and depression in the community, rather than rely on data from those who present in healthcare settings with mental health difficulties. However, it is important to emphasise that screening is not equal to a clinical diagnosis.

Due to the use of web surveys, which were accessible by mobile phone, a tablet, and computer [24], it was possible to start these surveys relatively fast after the outbreak of the pandemic. Another advantage of using a web survey is that it is very user-friendly, with high-quality data that are readily available [72,73]. However, it is important to note that online surveys also bring some disadvantages. The use of an online survey may have impaired the representativeness of the sample, with adults who cannot read, who cannot afford a computer/internet, and those who are less comfortable using a computer being potentially underrepresented [74].

The design of the online study also makes it difficult to reach some vulnerable groups in the population such as foreigners living in Belgium. However, survey findings indicated that the mental health of refugees and migrants during the COVID-19 pandemic was significantly impacted, particularly for certain subgroups (i.e., insecure housing situation and residence status) who reported experiencing higher levels of increased discrimination and increases in daily life stressors [75]. To target vulnerable groups, such as less educated people or people of foreign origin, it is better to opt for other data collection techniques such as interview-driven data collection [76].

We also cannot dismiss the possibility of a selection bias in our study: people who participated in all six waves of the study (15 months in total) may have experienced less or more anxiety and depression, compared with the people who drop out after one wave. However, because we performed a longitudinal study with the same participants over time, we were able to have a clear view of the effect of the different time periods with additional measures on mental health and contributing factors of mental health.

## 5. Conclusions

In summary, our findings suggest that the levels of anxiety and depression of a substantial proportion of the population may have been affected, especially during periods of strict measures imposed by the government. Evidence was found of coping and adjustment after the initial stress of the pandemic, as the proportion of participants with symptoms of anxiety and depression decreased during times of lower policy restrictions, to almost the same level as the pre-COVID times (BHIS2018) (especially for anxiety). This pattern of ‘resilience’ is commonly observed in response to stressful or traumatic life events [77].

As the COVID-19 pandemic is currently predicted to linger for an extended period of time, possibly even a few years [78], the governments need to develop methods to help those in the greatest need, especially during periods of strict lockdown. Risk groups should be targeted for counselling and extra social support. Especially, the increased risk of developing anxiety and depression among younger adults is concerning, as this is a group who may be experiencing mental health difficulties for the first time and, therefore, in need of early intervention. Previous research has established the substantial lifetime economic costs of mental health problems (e.g., through sickness absence, job loss, etc.) [52,79]. Therefore, investment in mental health treatment programs and supports is crucial.

## Figures and Tables

**Figure 1 behavsci-12-00141-f001:**
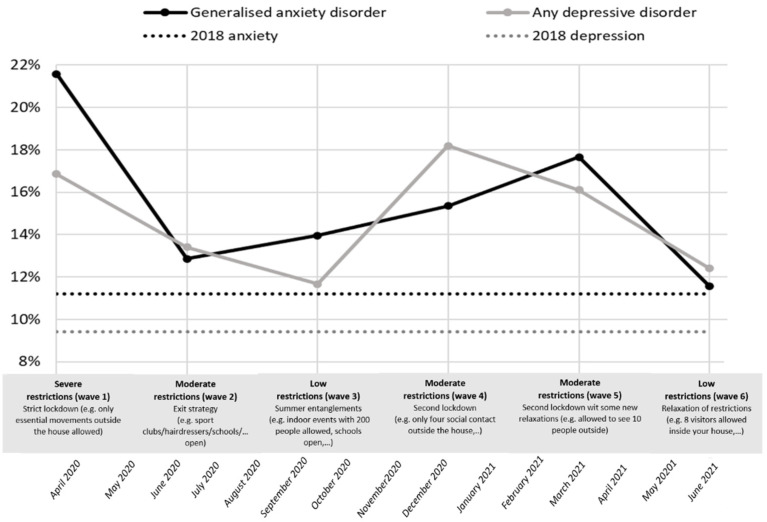
Evolution of anxiety disorder and depressive disorder during COVID-19 times (6 different time points)—weighted proportions (reference line: indication anxiety and depression, HIS 2018).

**Table 1 behavsci-12-00141-t001:** Different time periods (1 to 6) of the COVID-19 survey, number of respondents, level of restrictions, and examples of implemented restrictions.

T1	2–9 April 2020	N = 49,335	Severe restrictions	Strict lockdown (e.g., only essential movements outside the house allowed)
T2	28 May–5 June 2020	N = 33,913	Moderate restrictions	Exit strategy (e.g., sports clubs/hairdressers/schools/etc. open)
T3	24 September–2 October 2020	N = 30,845	Low restrictions	Summer entanglements (e.g., indoor events with 200 people allowed, etc.)
T4	3–11 December 2020	N = 29,855	Moderate restrictions	Second lockdown (e.g., only four social contacts outside the house allowed, etc.)
T5	18–25 March 2021	N = 20,410	Moderate restrictions	Second lockdown with some relaxations (e.g., allowed to see 10 people outside, etc.)
T6	10–20 June 2021	N = 17,774	Low restrictions	Relaxation of restrictions (e.g., 8 visitors inside your house allowed, etc.)

**Table 2 behavsci-12-00141-t002:** Descriptive analysis of the longitudinal data: people who participated in all 6 waves (crude %).

		T1		T2		T3		T4		T5		T6	
		N	%	N	%	N	%	N	%	N	%	N	%
Gender	Men	675	36.7	672	36.5	675	36.7	671	36.6	672	36.6	671	36.5
	Women	1162	63.2	1165	63.4	1160	63.2	1164	63.4	1164	63.3	1165	63.4
	Other	1	0.1	1	0.1	1	0.1	1	0.1	2	0.1	2	0.1
Region	Flanders	1182	64.3	1181	64.3	1176	64.1	1178	64.2	1180	64.2	1178	64.2
	Brussels	199	10.8	201	10.9	203	11.1	203	11.1	202	11.0	204	11.0
	Walloon	457	24.9	456	24.8	455	24.8	455	24.8	456	24.8	456	24.8
Age	18–29	94	5.1	91	5.2	96	5.2	94	5.1	66	3.6	68	3.6
	30–49	591	32.2	591	32.1	588	32.2	592	32.2	584	31.8	585	31.8
	50–64	682	37.1	682	37.1	682	37.1	680	37.0	659	35.9	657	35.8
	65+	471	25.6	470	25.6	469	25.6	470	25.6	529	28.8	528	28.8
Average age ± standard deviation			45.8 ± 14.2		49.9 ± 14.4		50.8 ± 14		51.4 ± 14.2		53.2 ± 14.1		52.7 ± 14.7
Household composition	Living alone, without children	389	21.2	399	21.7	404	22.0	411	22.4	414	22.5	417	22.7
	Couple, without child(ren)	748	40.7	746	40.6	738	40.2	749	40.8	746	40.6	753	41.0
	Couple, with child(ren)	497	27.0	499	27.2	512	27.8	495	27.0	499	27.1	485	26.4
	Living alone, with children	93	5.1	92	5.0	89	4.9	92	5.0	91	5.0	92	5.0
	Together with parent(s), family, friends or acquaintances	82	4.5	84	4.6	76	4.1	75	4.1	72	3.9	74	4.0
	Other	29	1.6	18	1.0	17	0.9	14	0.8	16	0.9	17	0.9
Work status	Paid job	1128	61.4	1125	61.2	1099	59.8	1097	59.7	1091	59.4	1089	59.4
	Unemployed (not temporarily interrupted))	43	2.3	43	2.3	40	2.2	35	1.9	40	2.2	40	2.1
	Invalidity	47	2.6	47	2.6	45	2.5	44	2.4	46	2.5	46	2.5
	Studies	16	0.9	17	0.9	16	0.9	15	0.8	14	0.8	14	0.7
	Retirement	534	29.0	549	29.9	570	31.1	577	31.4	590	32.1	595	32.4
	Household work	43	2.3	43	2.3	45	2.4	43	2.3	40	2.2	40	2.1
	Other	27	1.5	14	0.8	20	1.1	25	1.4	16	0.9	13	0.7
Education	Secondary degree (or lower)	477	25.9	476	25.8	481	26.2	474	25.8	475	25.8	479	26
	Higher education	1361	74.1	1362	74.2	1355	73.8	1362	74.2	1363	74.2	1359	74
Infected/Exposed to COVID-19	Yes	1364	74.2	1362	74.2	1363	74.2	1362	74.2	1364	74.2	1362	74.2
	No	474	25.8	473	25.8	474	25.8	474	25.8	474	25.8	475	25.8
Social Support													
	Poor	479	26	486	26.4	425	23.1	628	34.2	548	29.8	399	21.7
	Moderate	940	51.2	875	47.7	920	50.2	853	46.4	904	49.2	855	46.6
	Strong	419	22.8	476	25.9	490	26.7	354	19.3	386	21.0	584	31.8
Personality traits	Introversion	239	13.0	239	13.0	238	13.0	239	13.0	239	13.0	240	13.0
	Middle	1288	70.1	1288	70.1	1288	70.1	1286	70.1	1288	70.1	1286	70.1
	Extraversion	311	16.9	311	16.9	310	16.9	311	16.9	311	16.9	312	16.9
Pre-existing mental health problem	Yes	189	10.3	189	10.3	189	10.2	189	10.3	189	10.3	191	10.3
	No	1648	89.7	1649	89.7	1647	89.8	1647	89.7	1648	89.7	1647	89.7

**Table 3 behavsci-12-00141-t003:** Estimates (SE) from linear mixed models examining the total score on the GAD-7 anxiety scale (N = 1835).

		Model 1 (Intercept Only)	Model 2 (Time)	Model 3 (Time + Background Variables)	Model 4 (Other Explanatory Variables)	Model 5 (Time * Gender)
		Estimate (SE)	Estimate (SE)	Estimate (SE)	Estimate (SE)	Estimate (SE)
Fixed effects						
Intercept		4.16 *** (1.00)	3.89 *** (0.11)	2.37 *** (0.28)	0.77 *** (0.37)	0.83 * (0.37)
Time	1		1.17 *** (0.09)	1.16 *** (0.09)	1.12 *** (0.09)	0.75 *** (0.15)
	2		−0.16 (0.09)	−0.17 (0.09)	−0.20 * (0.09)	−0.08 (0.15)
	3 (ref.)					
	4		0.55 *** (0.09)	0.56 *** (0.09)	0.43 *** (0.09)	0.35 * (0.15)
	5		0.36 *** (0.09)	0.40 *** (0.09)	0.31 *** (0.09)	0.33 * (0.15)
	6		−0.26 ** (0.09)	−0.21 * (0.09)	−0.18 * (0.09)	−0.23 (0.15)
Gender	Man (ref.)					
	Woman			0.84 *** (0.18)	0.79 *** (0.17)	0.69 *** (0.21)
Region	Flanders (ref.)					
	Brussels			0.92 ** (0.29)	0.77 * (0.27)	0.77 ** (0.27)
	Walloon			0.95 *** (0.21)	0.90 *** (0.20)	0.90 *** (0.20)
Age	18–29 (ref.)			1.50 *** (0.40)	1.31 ** (0.39)	1.32 ** (0.39)
	30–49			1.38 *** (0.26)	1.07 *** (0.26)	1.08 *** (0.26)
	50–64			0.48 * (0.21)	0.29 (0.21)	0.30 (0.21)
	65+			-	-	-
Household type	Living alone, without children			−0.06 (0.20)	−0.28 (0.19)	−0.28 (0.19)
	Couple, without child(ren) (ref.)			-	-	-
	Couple, with child(ren)			0.05(0.18)	0.04(0.18)	0.03(0.17)
	Living alone, with children			−0.06 (0.32)	−0.32 (0.31)	−0.33 (0.31)
	Together with parent(s), family, friends, or acquaintances			−0.22 (0.34)	−0.28 (0.33)	−0.31 (0.33)
	Other			−0.59 (0.40)	−0.71 (0.39)	−0.70 (0.39)
Work status	Paid job (ref.)			-	-	-
	Unemployed (not temporarily interrupted))			0.82 * (0.32)	0.70 * (0.32)	0.70 * (0.32)
	Invalidity			0.85 * (0.39)	0.39 (0.38)	0.40 (0.38)
	Studies			−0.12 (0.60)	0.00 (0.59)	−0.01 (0.59)
	Retirement			−0.70 *** (0.20)	−0.68 *** (0.20)	−0.68 *** (0.20)
	Household work			0.31 (0.41)	0.39 (0.41)	0.40 (0.41)
	Other			0.17 (0.38)	0.17 (0.37)	0.16 (0.37)
Education	Secondary degree (or lower)			0.51 ** (0.17)	0.45 ** (0.17)	0.44 ** (0.17)
	Higher degree (ref.)					
Infected/exposed to COVID−19	Yes				0.47 ** (0.19)	0.47 ** (0.19)
	No (ref.)					
Social support	Poor				1.32 *** (0.12)	1.32 *** (0.12)
	Moderate				0.44 *** (0.09)	0.45 *** (0.09)
	Strong (ref.)					
Personality traits	Introvert (ref.)				-	
	Middle				0.64 * (0.25)	0.64 * (0.25)
	Extravert				1.98 *** (0.31)	1.98 *** (0.32)
Pre-existing mental health problem	Yes				2.61 *** (0.29)	2.60 *** (0.29)
	No (ref.)					
Time*Gender	1-Women					0.57 ** (0.19)
	1-Men (ref.)					
	2-Women					−0.19 (0.19)
	2-Men (ref.)					
	4-Women					0.12 (0.19)
	4-Men (ref.)					
	5-Women					−0.03 (0.19)
	5-Men (ref.)					
	6-Women					0.08 (0.19)
	6-Men (ref.)					
	3-Women					
	3-Men (ref.)					
Error variance						
Level-2		7.70 *** (0.11)	7.41 *** (0.11)	7.41 *** (0.11)	7.38 *** (0.11)	7.37 *** (0.11)
Level-1 intercept		14.71 *** (0.53)	14.76 *** (0.53)	13.18 *** (0.47)	11.30 *** (0.42)	11.30 *** (0.42)
*Model Fit*						
AIC		58,344	58,004	57,849	57,565	57,556
BIC		58,360	58,048	57,993	57,741	57,760

* *p* < 0.05; ** *p* < 0.01; *** *p* < 0.001. Values based on SAS PROC Mixed. Entries show parameter estimates with standard errors in parentheses. Estimation method = ML; Satterthwaite degrees of freedom.

**Table 4 behavsci-12-00141-t004:** Estimates (SE) from linear mixed models examining the total score on the PHQ-9 depression scale (N = 1835).

		Model 1 (Intercept Only)	Model 2 (Time)	Model 3 (Time + Background Variables)	Model 4 (Other Explanatory Variables)
Fixed effects					
Intercept		4.18 *** (0.10)	3.84 *** (0.11)	2.15 *** (0.28)	0.31 (0.38)
Time	1		0.44 *** (0.09)	0.43 *** (0.09)	0.38 *** (0.09)
	2		0.10 (0.09)	0.09 (0.09)	0.06 (0.09)
	3 (ref.)				
	4		0.77 *** (0.09)	0.77 *** (0.09)	0.62 *** (0.09)
	5		0.73 *** (0.09)	0.78 *** (0.09)	0.68 *** (0.09)
	6		0.05 (0.09)	0.11 (0.09)	0.14 (0.09)
Gender	Man (ref.)				
	Woman			0.57 ** (0.19)	0.49 ** (0.18)
Region	Flanders (ref.)				
	Brussels			1.18 *** (0.30)	1.05 *** (0.28)
	Walloon			1.09 *** (0.22)	1.06 *** (0.21)
Age	18–29			1.80 ***(0.40)	1.61 *** (0.38)
	30–49			1.43 *** (0.27)	1.11 *** (0.26)
	50–64			0.71 *** (0.21)	0.51 * (0.21)
	65+ (ref.)			-	-
Household type	Living alone, without children			0.56 *** (0.21)	0.34 (0.20)
	Couple, without child(ren) (ref.)			-	-
	Couple, with child(ren)			0.00 (0.18)	−0.05 (0.18)
	Living alone, with children			0.11 (0.32)	−0.13 (0.31)
	Together with parent(s), family, friends or acquaintances			0.52 (0.34)	0.41 (0.33)
	Other			−0.39 (0.39)	−0.53 (0.38)
Work status	Paid job (ref.)				
	Unemployed (not temporarily interrupted))			0.81 ** (0.31)	0.72 * (0.31)
	Invalidity			1.35 *** (0.39)	0.93 * (0.38)
	Studies			−0.39 (0.58)	−0.22 (0.57)
	Retirement			−0.67 *** (0.20)	−0.64 ** (0.19)
	Household work			0.68 (0.42)	0.72 (0.40)
	Other			0.38 (0.36)	0.40 (0.35)
Education	Secondary degree (or lower)			0.38 * (0.18)	0.31 (0.17)
	Higher degree (ref.)			-	-
Infected/exposed to COVID-19	Yes				0.43 * (0.21)
	No (ref.)				-
Social support	Poor				1.49 *** (0.11)
	Moderate				0.50 *** (0.09)
	Strong (ref.)				
Personality trait	Introvert (ref.)				-
	Middle				0.75 ** (0.27)
	Extravert				2.71 *** (0.33)
Pre-existing mental health problem	Yes				2.95 *** (0.30)
	No (ref.)				-
Error variance					
Level-2		6.76 *** (0.10)	6.64 *** (0.10)	6.64 *** (0.10)	6.59 *** (0.10)
Level-1 intercept		17.34 *** (0.61)	17.35 *** (0.61)	15.46 *** (0.55)	12.79 *** (0.47)
Model Fit					
AIC		57,415	57,261	57,102	56,710
BIC		57,431	57,306	57,245	56,886

* *p* < 0.05; ** *p* < 0.01; *** *p* < 0.001 Values based on SAS PROC Mixed. Entries show parameter estimates with standard errors in parentheses. Estimation method = ML; Satterthwaite degrees of freedom.

## Data Availability

The data of this study are available from the corresponding author upon reasonable request.

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
