# Peer review of "Anxiety and Depression in Belgium during the First 15 Months of the COVID-19 Pandemic: A Longitudinal Study"

_behavsci, 2022, doi:10.3390/bs12050141_

Round 1
Reviewer 1 Report
Title: The authors wrote about mental health, but, along the manuscript they focused the attention to anxiety and depression. I advise to change or adapt the title, adding specific terms (anxiety and depression). It should be useful also in the case of meta-analyses and systematic review.
At the end of the introduction, the authors stated the aims of the study. Given Belgium is one of the most multicultural country in Europe, did the authors include immigrants or foreign citizen with authorization to live in the country? I believe that is important to consider, given the impact of the COVID lockdowns on the mental health of immigrants or foreign (UE, extra UE citizens).
The methodology used in the manuscript is well written, clear and elegant.
I advise to use the term “depressive symptoms” instead of depression. The term depression implies a diagnosis.
In the introduction, the authors described previously published studies about the impact of the COVID-19 pandemic and restrictions on the mental health. This description highlighted the impact of COVID-19 not only on the single subject but also at social levels. Indeed, the authors cited several previous published studies that investigated the relationship between COVID-19 and mental health in other countries (i.e. UK), and international (WHO). The authors also reported and cited a systematic review. Moreover, they underlined the role of several factors, such as “closure of schools, universities and support services and in-49 creased pressure on families” that can affect socialization and coping strategies to deal with daily stressful situations. Despite this, the authors introduced their study also adding a specific sentence about the prodromal factors ( preexisting mental health problem) of distress during a crisis ( as you can read: “ Furthermore, having pre-existing mental health problems is associated with higher levels of psychological distress during this crisis 54 [17]”. Therefore, the authors pointed to personality factors (introversion vs extraversion) that can be affected in a different way by the COVID-19 restriction (Line 62 p.2). Indeed the explanation given by the authors is not detailed about the relationship between a societal crises and Introversion vs extraversion. I advise the authors to explain better the impact of crises on the subjects with introverted or extraverted traits or a prevalence of them (indeed the found “middle extravert”). The aim, as stated by the authors is well delineated, introducing in a more specific way the specific hypotheses, representing a good way to address the research problem from the general aims to the specific operationalized hypotheses.
In the MATERIALS and METHODS, they reported in a detailed way, the specific setting of restriction in Belgium adding a table with the timeline of the restrictions in Belgium associating the number of responders to the surveys (Table 1). In the same way, The authors reported the dataset composed by a national survey from Sciensano ( Belgium authority), reporting the number of waves (I prefer time-points : T1, T2 … Tx…. Tn) . In this way, authors should review the text and adopt a more specific language. Reported the number of responders (T1-T69, the software (Lymesurvey) and also previous survey for a normative comparison. They described the the psychometric test (self-administered) with the psychometric properties highliting test –retest stability (line 155 p.4). I agree with the use of tests with few items (GAD-7 , PHQ -9 BIF 2 S) to limit the number of non-responders. Maybe this should be added to the text. The authors also added the predictor variables. According to me this part should written in a more schematic way and specifying better the reason for the use of each predictor. The statistical analysis has been explained in detailed way, allowing the opportunity to replicate the same analysis for a (different) longitudinal data from survey. Indeed, authors reported in supplementary info the items used in the survey (important for replication as I above mentioned) and the linear model built for the analysis. Moreover, they explained and described all the test used in the analysis motivating, with refs, their application in the study. This is useful also for students and fellows that can apply a similar analytic model to future studies. I found very interesting the effect of predictor on the model 4 that the authors explained and motivated in the discussion. However, the discussion should be written at the light of a frame-of-reference model (for example the mental health and economic crisis or in war refugee) or but I agree with the authors’ cut discussing the result for each social group. Moreover, I agree with the limitation of study.
Reviewer 2 Report
This is a very intersting studies presented in a straight forward written manuscript. It is an important contribution to the recent literature.
Reviewer 3 Report
There are lots of researches focusing the psychological, mental, social effects of COVID 19 on people's life, however, this study comes forward as it was designed as a longitudinal study. Longitudinal studies are valuable sources to see how the particpants perceptions change in time. Thus I read the paper with a great interest.
In general the study is well-designed and the methodology is fine . There are some minor points that should be fixed and I think the issues that I raised in this review needs minor revision and they will strenghten the research.
I think the second hypotheses is too long and I recommend the authors to rewrite (if possible). Another point in the paper is that there are some paragraphs which contains only one sentence (see page 4, line 156, page 11, line 323 etc.). A paragraph is a set of sentences and I recommend the authors to avoid using paragraphs which have only one sentence. These sentences may be combined with the previous or next paragraphs.
Another importnat point is that I could not see any explanations after the tables. The authors can add short explanations after the tables.
And my last recommendation is about the disscussion part. The authors can develop their disscussions by citing international literature. There are lots of studies on the psychological effects of COVID-19 from Italian, Spanish, German, Turkish context.
Regards
